# Recent Advances in Structural Studies of Single-Stranded RNA Bacteriophages

**DOI:** 10.3390/v15101985

**Published:** 2023-09-23

**Authors:** Jirapat Thongchol, Zachary Lill, Zachary Hoover, Junjie Zhang

**Affiliations:** Center for Phage Technology, Department of Biochemistry and Biophysics, Texas A&M University, College Station, TX 77843, USA; jirapat.tho@tamu.edu (J.T.); zachlill@tamu.edu (Z.L.); zachhoov@tamu.edu (Z.H.)

**Keywords:** ssRNA phages, adsorption, replication, packaging, single-lysis protein

## Abstract

Positive-sense single-stranded RNA (ssRNA) bacteriophages (phages) were first isolated six decades ago. Since then, extensive research has been conducted on these ssRNA phages, particularly those infecting *E. coli*. With small genomes of typically 3–4 kb that usually encode four essential proteins, ssRNA phages employ a straightforward infectious cycle involving host adsorption, genome entry, genome replication, phage assembly, and host lysis. Recent advancements in metagenomics and transcriptomics have led to the identification of ~65,000 sequences from ssRNA phages, expanding our understanding of their prevalence and potential hosts. This review article illuminates significant investigations into ssRNA phages, with a focal point on their structural aspects, providing insights into the various stages of their infectious cycle.

## 1. Introduction

Positive-sense ssRNA phages are genetically and structurally simple RNA viruses that infect Gram-negative bacteria by utilizing bacterial retractile pili as receptors. The discovery of the first ssRNA phages occurred in 1960 [1], specifically those infecting *E. coli* through the F-pilus. Subsequently, other ssRNA phages that rely on different types of retractile pili as receptors have been isolated. Up until 2016, only a limited number of ssRNA phage species had been documented in databases. Table 1 provides an overview of the host range and receptors for those known ssRNA phages, highlighting their ability to evolve and target various types of retractile pili, such as the conjugative type IV secretion system (T4SS) pilus, the type IV pilus (T4P), and the tight adhesion (Tad) pilus.

ssRNA phages are lytic phages characterized by small genomes of approximately 3000–4000 nucleotides. These genomes usually encode four essential proteins: the maturation protein (Mat), the coat protein (CP), the β-subunit of the replicase (Rep), and a single lysis protein (Lys) (Figure 1A). Mat and CP are responsible for the phage’s structural integrity, while Rep replicates the viral genome within the host cell. Lys is necessary for lysing the host cell, allowing the release of newly formed viral progeny into the environment. The genomic architecture of ssRNA phages typically follows a similar pattern from the 5′ to the 3′ ends, with the *mat* gene located at the beginning, followed by *cp* and *rep*. However, the *lys* gene is an exception. It seems to have evolved to be distributed throughout the genomes of different ssRNA phages.

In 2016, ssRNA phage research entered the metagenomic and transcriptomic era, with 158 ssRNA phage sequences being identified [2]. From 2018 on, ssRNA sequences increased drastically, leading to the identification of thousands of new ssRNA genomes [3,4,5]. The identification of these phage genomes from metagenomic data of environmental samples is based on the detection of the *rep* gene sequence, which is highly conserved among ssRNA phages. This breakthrough prompted the taxonomic reclassification of ssRNA phages, resulting in the creation of two new orders (based on the Rep) and six families (based on the CP), into a total of 428 genera (50% Rep pairwise amino acid sequence identity) and 882 species (80% Rep pairwise amino acid sequence identity) [6]. This not only highlights the prevalence of ssRNA phages in nature, but also suggests the potential existence of additional hosts and retractile pili that ssRNA phages can target. However, the hosts of these newly identified ssRNA phages remain unknown. By 2022, the number of identified ssRNA phage genomes had further expanded to 65,814 sequences [4]. Using the Hidden Markov Model (HMM), which is a statistical model to analyze protein sequence similarity and identify protein domains, from the 2020 literature [3], we revealed more possibly near-complete genomes (~12,288 genomes) in the 2022 dataset [4] that possess the three core genes: *mat*, *cp*, and *rep*.

**Table 1 viruses-15-01985-t001:** Examples of host ranges and receptors of previously known ssRNA phages [7,8,9,10,11,12,13,14,15,16].

Phages	Genus	Hosts	Receptors
MS2	*Emesvirus*	*E. coli*	Conjugative F pili
Qβ	*Qubevirus*	*E. coli*	Conjugative F pili
PP7	*Pepevirus*	*P. aeruginosa*	Type IV pili
LeviOr01	*Pepevirus*	*P. aeruginosa*	Type IV pili
AP205	*Apeevirus*	*Acinetobacter* spp.	Type IV pili
ϕCb5	*Cebevirus*	*C. crescentus*	Type IV Tad pili
PRR1	*Perrunavirus*	*Pseudomonas*, *Salmonella*, *Vibrio*, *Escherichia*	Conjugative P pili
M	*Empivirus*	*Escherichia*, *Salmonella*, *Klebsiella*, *Proteus* and *Serratia*	Conjugative M pili
C-1	*Cunavirus*	*Escherichia*, *Salmonella*, *Proteus* and *Serratia*	Conjugative C pili
Hgal1	*Hagavirus*	*Escherichia*, *Citrobacter*, *Klebsiella*, *Enterobacter*	Conjugative H pili

The infection cycle of ssRNA phages commences with their adsorption to the host retractile pilus by the Mat (Figure 1B). After adsorption, the genomic RNA (gRNA) of the ssRNA phage enters the host cell through an unknown mechanism. Inside the host, the positive-sense ssRNA genome acts as mRNA and is translated by host ribosomes, producing the phage-encoded proteins. The expressed Rep hijacks and assembles with host proteins, which include the ribosomal protein S1 and the elongation factors Tu and Ts (EF-Tu and EF-Ts), forming a holoenzyme that drives the synthesis of additional gRNA [17,18]. This process amplifies the translation of phage proteins and production of more gRNA. The newly synthesized phage capsid proteins (Mat and CP) and gRNA undergo assembly to form mature virions. Cell lysis is then achieved by the lysis protein to release the newly assembled ssRNA virions.

In this review article, we highlight recent studies of ssRNA phages, covering these infection steps, with a particular focus on their related structures.

## 2. The Known Structures of ssRNA Phage Capsids: VLPs and Mature Virions

Prior to the metagenomic era of ssRNA phage, extensive studies were conducted on the capsid structures of known ssRNA phages using both X-ray crystallography and cryo-electron microscopy (cryo-EM) (Appendix A). These structures, known as virus-like particles (VLPs), were solved as non-infectious forms, mostly through the overexpression of CPs, which form capsids with an icosahedral symmetry (triangulation number, T = 3). Later, Rumnieks J. et al. overexpressed and solved 22 VLP structures of pre-2018 unculturable ssRNA phages [19]. Some 10 out of the 22 phages had CP structures that consisted of the typical MS2 CP fold, while the remaining phages had substantial differences in the CP, especially in the N-terminal region. This work revealed deviations in wild-type CPs of ssRNA phages, suggesting classification of CP proteins into different groups. After the arrival of new metagenomic data, the CPs were eventually classified into eight distinct groups: A through H [3,6].

While these structures offer valuable insight into the fold of the CP and the symmetry of the capsids, they do not provide information regarding how the gRNA is packaged, as well as how Mat proteins are assembled in the mature virion. Not until 2016 (~55 years since the discovery of ssRNA phages) were the mature structures of *E. coli* ssRNA phages (coliphages) reported at high resolutions. These F-specific coliphages are MS2 [20] and Qβ [21]. 

The cryo-EM structures of MS2 have provided insights into the composition of mature ssRNA phage virions [20,22,23]. These virions consist of a capsid that encapsidates a single strand of gRNA, along with a single copy of Mat. The gRNA is folded into a defined tertiary structure inside the virion capsid. The capsids are near-icosahedral, with 89 CP dimers and the Mat disrupting the perfect T = 3 icosahedral symmetry at one of the 2-fold axes of the particle (Figure 2A).

Given the similar genome architectures and high sequence similarity among different CPs, it seemed reasonable to speculate that other ssRNA phages might possess a comparable structure to that of MS2. While the cryo-EM structure of mature Qβ exhibits similarities to MS2, it also varies from MS2 [21]. Surprisingly, Qβ contained 90 copies of CP dimers, with 89 forming the capsid and one becoming encapsidated alongside the gRNA (an internalized CP dimer) [24] (Figure 2B). 

Mat is a crucial protein that plays a role in determining the maturity of ssRNA phages [25]. Despite its significant role in binding to gRNA and the host receptor, limited structural information is available for the Mat due to the challenges of purifying it; Mat proteins tend to be insoluble when overexpressed [26]. So far, the structures of Mat proteins have only been elucidated for ssRNA coliphage MS2 and Qβ. Due to the difficulties in the purification of Mat alone, cryo-EM has proven to be a powerful tool for exploring the structure of Mat within mature virions. The structure of Mat^MS2^ was first successfully revealed using cryo-EM [20]. Mat^Qβ^, also known as A_2_, was the first successfully purified Mat in vitro through an MBP-tagged approach [27], and was later characterized using X-ray crystallography [28]. 

**Figure 2 viruses-15-01985-f002:**
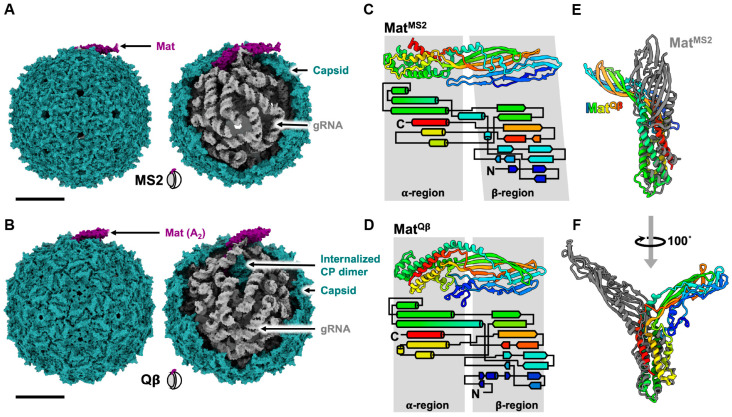
The known structures of F-specific ssRNA coliphage MS2 and Qβ. (**A**) The atomic model of MS2 (capsid combined from PDB 6NM5 and 5MSF, and RNA modeled using conformation 1 from Chang et al. [29]). The scale bar denotes 100 Å. (**B**) The atomic model of Qβ (from PDB 7LHD). The scale bar denotes 100 Å. (**C**) The ribbon model and secondary structural topology of Mat^MS2^. The tip of the β-sheet, colored blue and cyan, only has a backbone model in PDB 6NM5. Therefore, we used AlphaFold to predict the secondary structures and sidechains of this region (**D**) The ribbon model and secondary structural topology of Mat^Qβ^. (**E**) The structures of Mat^MS2^ and Mat^Qβ^ aligned through α-helical region. (**F**) A 100-degree rotational turn from that of panel (**E**).

Mat^Qβ^ (NCBI_001890) and Mat^MS2^ (NCBI_001417) share ~19% sequence identity, ~30% sequence similarity, and fold into similar tertiary structures comprised of an α-helical region and a β-sheet region (Figure 2C,D). However, these two Mat proteins display two major differences. First, Mat^Qβ^ serves an additional function, distinct from that of MS2, as a lysis protein. Second, Mat^Qβ^ exhibits a structural perturbation in comparison to Mat^MS2^ (Figure 2E,F): When aligning α-helical regions, the β-sheet regions of Mat^Qβ^ and Mat^MS2^ are oriented in opposite directions. It is worth noting that both MS2 and Qβ infect the same host through the same receptor. 

Notably, the proportions of mature virions containing Mat proteins and gRNA in the purified particles vary significantly between MS2 and Qβ. According to the published cryo-EM structures, MS2 efficiently assembles ~85–90% viral particles into mature virions, while Qβ only exhibited ~25–30% mature virions, a ~3 fold lower Mat incorporation than MS2 (Table 2). This might explain the observation that Qβ adsorbs to the F-pilus less frequently than MS2 [30]. 

## 3. The Interactions between Host Receptors and ssRNA Phages

Early studies showed that ssRNA phages readily adsorb to the side of their host pilus [32,33]. However, there is debate as to whether pilus binding alone is sufficient to trigger the gRNA release from the phage particle [34,35]. With the emergence of cryo-EM, the pioneering structural investigation of the interaction between ssRNA phages and their host receptors focused on the MS2 and F-pili pair [31,36]. The structure of the MS2/F-pilus complex shows a complete MS2 virion with intact gRNA inside, indicating the gRNA has not yet been released upon pilus binding. 

In 2013, Dent et al. employed cryo-electron tomography (cryo-ET) and sub-tomogram averaging to explore the MS2/F-pilus complex, which resulted in an asymmetric reconstruction of the complex at an approximate resolution of 39 Å [36]. While this density map did not offer detailed information on residue–residue interactions, it revealed additional density, which may correspond to the Mat protein, at the binding interface. This observation provided supporting evidence that the Mat disrupts the perfect icosahedral T = 3 capsid symmetry and is crucial for F-pilus binding.

Six years later (2019), Meng et al. employed single-particle cryo-EM to unravel the structural basis of MS2′s interaction with its receptor, the F-pilus, at a resolution of ~5–7 Å [31]. This study demonstrated that the Mat^MS2^ utilizes its β-sheet region to interact with four F-pilin subunits spanning two adjacent helical turns (Figure 3A,B). The interface area of the interaction was approximately 1063 Å^2^ (Figure 3B). The binding between Mat^MS2^ and the pilins involved electrostatic and hydrophobic interactions (Figure 3C,E). The first mode of electrostatic interaction occurred between Mat residue R99 and pilin chain A residue D23 (Figure 3C). The second mode involved Mat residue R36 and pilin chain C residue D7 (Figure 3D). Additionally, the hydrophobic residues F94 and F92 of Mat interacted with the N-terminus of pilin chain D (Figure 3E). The binding of MS2 to the F-pilus occurred in a specific orientation, with Mat pointing its tip away from the cell surface. This orientation is proposed to facilitate gRNA entry upon retraction, especially when brought closer to the basal body of the Type IV secretion system [31].

Unexpectedly, in 2020, Harb et al. demonstrated that both MS2 and Qβ trigger detachment of F-pili during entry, as fragments of pili appeared in the media, and this occurs in a phage concentration-dependent manner [37]. Even in the absence of the Type IV coupling protein, TraD (a protein which is part of the bacterial Type IV secretion system and required for conjugation), the mutant T4SS F-pilus was still able to be detached by MS2. However, the gRNA of MS2 failed to enter the host cell’s cytoplasm [37]. Notably, TraD has been shown to be necessary for the MS2-related phages R17 and f2, but not Qβ [38]. The exact discrepancy in the requirement of TraD for gRNA entry and infection between MS2 and Qβ remains elusive. Although it is unknown how the gRNA of ssRNA phages enters the host cell, these results have emphasized the requirement of pilus retraction. Based on Harb et al.’s results, pilus detachment is necessary, and gRNA penetration into the host cell possibly starts once the phages have been brought close to the cell surface via retraction of the pilus assembly machinery. This mechanism seems to be conserved among ssRNA phages, since PP7 has also been shown to result in a 50% reduction in length of the type IV pilus during infection [32]. It is not clear what the exact mechanism of ssRNA phage-induced pilus detachment is. It may be that the first ssRNA phage entering the cell causes the pilus to detach, leaving the other pilus-adsorbed phages unable to infect the cell. This process resembles superinfection exclusion in other phages.

## 4. The Genome Replication of ssRNA Phages

The ssRNA phage *rep* gene encodes the β-subunit of the replicase (Rep), an RNA-dependent RNA polymerase, which contains the catalytic site responsible for replication. The replicase holoenzyme consists of the virally encoded β-subunit as well as three essential host translation proteins. They are the elongation factor thermal stable (EF-Ts), the elongation factor thermal unstable (EF-Tu), and the S1 protein of the ribosome (Figure 4A) [39]. These proteins play crucial roles in efficient translation during normal cellular function. EF-Tu is a G-protein that functions in translocating amino-acylated tRNAs, EF-Ts facilitates the exchange of GDP for GTP for EF-Tu, and the S1 protein promotes translation initiation by stabilizing the mRNA [40,41].

**β-subunit domain architecture:** The structural and enzymatic investigations of ssRNA replicase have primarily focused on the canonical Rep^Qβ^ due to its stability. Rep^Qβ^ has been shown to be expressed and purified successfully by fusing EF-Tu, EF-Ts, and Rep into a single chain connected by flexible linkers [42]. Structural studies reveal that Rep^Qβ^ contains three distinct domains: palm, thumb, and fingers. Additionally, Rep^Qβ^ possesses a bridge region that connects the finger and thumb domains and is important in complex formation (Figure 4B) [43,44]. The palm domain of Rep^Qβ^ contains the catalytic site, which coordinates two divalent cationic metal ions [45], typically calcium or magnesium, via three aspartic acid carboxylates for RNA polymerization. The two metal ions help coordinate the growing RNA strand and the next nucleotide. 

**Interactions with host proteins in the holoenzyme:** Within the replication complex, it has been determined that the OB1 and OB2 domains of the S1 protein interact with the finger domain (Figure 4A), but the OB3-OB6 domains are flexible and cannot be resolved [46]. In the Qβ holoenzyme, EF-Tu domain 2 has been identified as interacting with the finger domain of the β-subunit, while EF-Tu domain 3 and EF-Ts interact with the thumb domain of the β-subunit [43,44]. It is notable that the EF-Tu and EF-Ts proteins are known to form a complex that has been observed outside of the Qβ holoenzyme [40]. These observations indicate the complex nature of the interactions within the replication complex, highlighting the intricate interplay between the various components involved in RNA polymerization.

The HMM search on ssRNA phage sequences (~15,000 sequences) from the 2020 literature identified two distinct orders of ssRNA phage Rep proteins based on 70% pairwise amino acid identity [3]. In 2022, a larger collection of ssRNA sequences (~65,000 sequences) also showed consistency with two major lineages as seen in the phylogeny generated (Figure 5A) [4]. Although the catalytic core of Rep is conserved, the host–factor interacting region displays a relative diversity (Figure 5B), suggesting that they have evolved to bind to different host factors from their respective hosts.

**Initiation:** Replication is initiated at the 3′UTR of the positive-sense genome. It has been described that in Qβ the OB3 domain of the S1 protein is able to recognize and bind to two internal regions of RNA in the Qβ genome, 1247–1346, known as the S-site, and 2545–2867, known as the M-site [49]. This allows the complex to bind the RNA and position the CCA-3′ sequence in the active site. While A is the terminal residue, replication starts at the penultimate C. When the terminal residue is mutated from A to G, U, or deleted, there is a significant reduction in replication [50]. From these studies, the idea emerged that the non-template A functions in stabilizing the initiation complex through Pi-Pi interactions with the GTP and the penultimate C [45].

**Elongation:** The bases enter through the NTP channel and hydrogen bond with the current nucleotide on the template strand, forming an RNA duplex. As each base is added, the duplex is driven towards the EF-Tu subunit of the complex. Once the ninth nucleotide is added to the growing strand, the non-template 3′ adenosine Pi stacks onto a C-terminal asparagine of the β-subunit [45]. This wedge region, formed by the C-terminus and EF-Tu, begins to destabilize the duplex and is the basis for its separation. Upon addition of the tenth nucleotide, the non-template A “flips” around the wedge through its interactions with domains 2 and 3 of EF-Tu and begins the separation of the template and growing RNA strands. At nucleotide 14, the template RNA begins to leave the replication complex through the exit tunnel formed by the β-subunit-EF-Tu interface.

**Termination:** Once replication reaches the terminal 5′ end of the genome, termination commences. The final template base, C, is added to the growing negative sense RNA, and then this complex shifts, allowing for the NTP binding site to open. It has been shown that this binding pocket is too large for CTP and UTP to bind in and too small for GTP, but it is the correct size for ATP, which is preferentially incorporated in the 3′ growing strand [51]. In addition to the binding pocket size, the adenine base can Pi stack onto the 5′ template strand G stabilizing the RNA duplex. This is the proposed mechanism of non-template A addition. Furthermore, it is thought that the S1 protein aids in termination of replication through the binding of the growing RNA via the OB3 domain [52]. This prevents annealing of the template and growing strands upon release, but it has not currently been shown where the S1 protein binds on the negative sense RNA [53].

## 5. The Genome Packaging and Viral Assembly of ssRNA Phages

The genome packaging of ssRNA phages differs from dsDNA and dsRNA phages, in which ssRNA phages lack motors/effector proteins that help package the genome into the capsid [54,55]. ssRNA phages do not have a highly pressurized capsid. Instead, the gRNA of ssRNA phages is known to form a high level of secondary structures comprised of many RNA stem-loops, upon which the Mat and CPs will assemble to form a mature virion (Figure 6) [29,56]. The very first RNA stem-loop discovered was shown to bind with high affinity to CP dimers and was called “an operator”. This is the RNA stem-loop that contains the start codon of the *rep* gene and is shown to play a role in suppressing the expression of *rep.* The operator stem-loop also acts as a strong encapsidation site for the coat protein to bind. These operators are shown to be conserved among known ssRNA phages [57,58,59]. 

The operator-like stem-loops are found throughout the genome of ssRNA phages and are believed to act as “packaging signals” in ssRNA viruses [60]. Chang et al. 2022 modeled the entire gRNA of Qβ and identified a total of 77 RNA stem-loops (Figure 6A). Out of these, 59 stem-loops were identified to interact with the capsid within 5 Å distance (Figure 7A–E) [56]. Among these 59 RNA stem-loops in Qβ, 32 were identified as “operator-like” RNA stem-loops. These 33 stem-loops (32 operator-like and 1 actual translational operator, number 34) interact with the coat protein dimer in the same manner based on their model, and were proposed to be directly involved in the viral assembly (Figure 7D,E, red numbers). The remaining 26 RNA stem-loops were identified as non-operator-like stem-loops (Figure 7E, black numbers). Stem-loops 57 and 59 are those that interact with the internalized CP dimer and Mat^Qβ^, respectively (Figure 7D,E). In contrast, in the high-resolution cryo-EM structure of MS2 solved by Dai et al. 2017 (Figure 7F–J) [20], 14 RNA stem-loops, out of 71 in the MS2 gRNA, interacted with the mature virion capsid and were resolved to high resolutions, thanks to their high affinity for the CP shell (Figure 7I, black-outline circles). Chang et al. later on built a complete model of the MS2 gRNA [29], which allows us to identify five additional RNA stem-loops shown to be “operator-like” (Figure 7I, dotted-outline circles). This increases the total to 40 RNA stem-loops defined as stem-loops within less than 5 Å to the inner surface of the MS2 capsid (Figure 7I,J). Among those, 19 stem-loops were identified as operator-like stem-loops (Figure 7J, red numbers), while 21 stem-loops were identified as non-operator-like stem-loops (Figure 7J, black numbers). The actual MS2 translational operator is stem-loop 18, while stem-loop 40 interacts with Mat^MS2^ (Figure 7I,J). These results revealed the gRNA packing preference in mature virions. With the Mat defined as the “north pole”, in both MS2 and Qβ, the 5′ end of the genome resides towards the southern hemisphere of the capsid, while the 3′ end resides towards the northern hemisphere (Figure 7D,I). Notably, there is a crack in the Qβ mature capsid around where the Mat points [21], which suggests that this region of the capsid is the last to assemble around the RNA.

Besides the near-icosahedral T = 3 mature virion and perfect icosahedral T = 3 VLPs, it has been previously shown that ssRNA phage capsid proteins are capable of assembling into non-canonical T = 4 [19,61,62,63], T = 1 [64], elongated T = 3 Q = 4 (prolate) [19], unusually large T = 3 [19], and even tubular capsids [65] when overexpressed or mutated (Figure 8A and Appendix A; Table A1). Chang et al. demonstrated that Qβ was able to assemble into small populations of non-canonical capsid forms through wild-type infection [56]. The various non-canonical forms of Qβ reported include T = 4, prolate, oblate and small prolate capsids (Figure 8B). All these forms of Qβ contain 12 pentamers of CPs, but the number of hexamers changes in correlation with the size of capsid (Figure 8C). This suggests that the manipulation of the CP hexameric units incorporated into the particles might influence the size of Qβ capsid. In conjunction with this, the size of RNA might also influence the capsid assembly. Indeed, Chang et al. 2022 showed that when only *cp/read-through* were overexpressed, the population of oblate and small prolate particles increased [56]. 

The current proposed model of gRNA packaging and assembly in ssRNA phages may be referred to as co-replicational assembly [56]. This model describes that the packaging starts during the replication of negative-sense to positive-sense RNA. The 5′ end of the nascent positive-sense RNA folds into its secondary structure, exposing the operator-like RNA stem loops which act as a packaging signal, providing a nucleation site for CP dimers (referred to as C/C dimer). Upon binding to these packaging signals, the CP dimers change their conformation from C/C to A/B, which promotes pentamer formation. As replication progresses, CP recruitment continues, leading to RNA collapse. This forms the intermediate state, leading to the folding of tertiary structures of the RNA. Upon replication termination, the 3′ end of the RNA becomes exposed and recruits the Mat, resulting in a mature virion. 

## 6. The Lysis of the Host by ssRNA Phages

Double-stranded DNA (dsDNA) phages usually lyse their host cells through multi-gene lysis systems, which are typically composed of endolysins and holins. Endolysins are muralytic enzymes that target the peptidoglycan cell wall synthesis pathway, while holins are cytoplasmic membrane-spanning proteins that create “holes” on the membrane, allowing endolysins to access and degrade the peptidoglycan cell wall [66]. Unlike dsDNA phages, ssRNA phage host lysis is induced through a single-gene Lys [67]. Lys is encoded by the *lys* gene, and, unlike their other structural and functional genes, is widely distributed throughout the genome of ssRNA phages. Typically, it is embedded out of frame in other genes (Figure 1A). 

The Lys proteins of known ssRNA phages have been classified into two groups: (1) non-peptidoglycan (PG)-targeting and (2) PG-targeting Lys proteins (Figure 9A). The mechanism of non-PG-targeting Lys remains inscrutable. Unlike the latter group, as the name suggests, they do not target enzymes related to peptidoglycan biosynthesis. The canonical Lys^MS2^ is shown to have four domains [68]. Through comparative analysis of sequences, several candidates were identified as non-PG-targeting [68]. Interestingly, one of the culturable ssRNA phages targeting *P. aeruginosa*, LeviOr01, does not have a Lys. Although the *lys* candidate has been proposed, the start codon found in the annotation is an arbitrary 5′-TTA-3′ (which codes for leucine). Thus, it is likely that the annotation is not actually a *Lys.* LeviOr01 was shown to form clear plaques on *P. aeruginosa* PcyII-10. The sub-population of PcyII-10, after being passed for multiple rounds, can still produce LeviOr01. Hence, it is proposed by the authors that this ssRNA phage is capable of inducing a carrier state in PcyII-10 [10]. 

The second group of Lys proteins is the PG-targeting Lys, which targets enzymes involved in PG synthesis (Figure 9A). These PG-targeting Lys include Qβ, M, and PP7 [67]. Nonetheless, the structural information for ssRNA Lys proteins is only limited to Lys^Qβ^, also known as A_2_. A_2_, which also functions as Mat^Qβ^, promotes lysis by inhibiting MurA (UDP-N-acetylglucosamine enolpyruvyl transferase). This enzyme catalyzes the first committed step in the biosynthesis of PG. The binding of MurA to its substrate, UDP-N-acetylglucosamine (UDP-GlcNAc), causes a conformational change of MurA from an open to a closed conformation. The closed conformation of MurA allows A_2_ to bind to the UDP-GlcNAc-MurA complex, thus inhibiting MurA’s function (Figure 9B,C) [24,69]. Although it is lacking structural information, Lys^M^ and the recently identified Lys^PP7^ are shown to inhibit MurJ, an enzyme with lipid II flippase activity [70]. One of the key residues in *E. coli* MurJ that confers function of Lys^M^ and Lys^PP7^ is Q244. Studies have shown that the mutation of this residue to proline (Q224P) causes resistance to both Lys^M^ and Lys^PP7^. This residue is located on transmembrane domain (TMD) 7, one of the 14 TMDs of MurJ [70,71]. However, it is noteworthy that Lys^M^ and Lys^PP7^ are different in structure and sequence. In fact, sequence analysis of Lys^PP7^ suggests that it might be a non-PG-targeting Lys [68,71].

## 7. Conclusions and Future Directions

Even with the many years of research on ssRNA phages, current knowledge of the structures and biology of ssRNA phages still revolves around ssRNA coliphages (Figure 2), except for the structures of the CPs (Appendix A). However, recent advancements in metagenomes and metatranscriptomes [2,3,4,5] have unveiled a myriad of ssRNA phages that exceeded previous expectations. The mature virions of the ssRNA coliphage, MS2 and Qβ, were shown to contain a single copy of Mat and a capsid that was composed of 178 copies of CPs encapsidating gRNA. Despite infecting the same host and utilizing the same receptor as MS2, Qβ notably exhibits structural variations, containing an internalized CP dimer in its mature virion [22]. Currently, with around ten ssRNA phages that can be cultured, their diversity and variation have already been demonstrated. It is unknown if the mature virions/morphology observed in ssRNA coliphages will be valid for all culturable ssRNA phages, such as PP7, LeviOr01, ϕCb5, and AP205, since they have also evolved to target different types of retractile pili. The mature structures of these phages are worth exploring to understand the biology of ssRNA phages, especially the Mat, which is challenging to purify. Structural determination using cryo-EM is a promising strategy to unravel the structures of these ssRNA mature virions. 

With an increase in the number of ssRNA phage sequences, it is possible that the host range and retractile pilus receptors that these ssRNA phages target could be more diverse. Exploring the interaction of Mat/pilus unveils information necessary to understand the adsorption of these phages. With enough structural information, motif searches into metagenomic data could allow us to identify new receptors or new hosts for new ssRNA phages, particularly facilitated by the recent breakthrough in protein structure prediction tools such as AlphaFold2, RoseTTAFold, I-TASSER, and ESMFold [72,73,74,75]. It is worth noting that the entry process of these ssRNA phages remains unknown, particularly regarding the involvement of the Mat in coordinating gRNA delivery into the cell. Although it was shown in MS2 that pilus detachment is required for gRNA entry [37], it is unclear how gRNA translocates through the cell membrane/peptidoglycan into the host cell. This phenomenon could potentially be unraveled through cryo-electron tomography.

Although there are complete atomic models of gRNA for coliphage MS2 and Qβ and different forms of VLPs (Figure 2 and Figure 8, and Appendix A), which suggests a potential genome packaging pathway for ssRNA phages, the proposed model has yet to be tested or visualized in real time in situ. Using combinations of time-resolved cryo-EM [76] and focus-ion beam (FIB)/cryo-ET [77] could potentially unravel the gRNA-packaging mechanism both in vitro and in situ, and observe any phage assembly intermediates during gRNA packing. 

The model for gRNA replication by Rep for ssRNA phages is mainly derived from coliphage Qβ. It is reasonable to suggest that the gRNA replication carried out by replicase might function differently for different host targets. One explanation would be that the host factors involved in the process might differ for ssRNA phages targeting different hosts. It has been shown in Qβ that the host factor for Qβ replication (Hfq) modulates Rep^Qβ^ activity for negative-sense gRNA synthesis [78,79]. Recently, additional host factors for Rep^MS2^ have also been identified, including initiation factor-1 (IF-1) and IF-3. While IF-1 was shown to promote replication, IF-3 was shown to inhibit it [80]. The structural interactions of these additional host factors with the holoenzyme replicase/RNA complex are yet to be explored. 

Lysis is the terminal step of the ssRNA infectious cycle taken to release new viral particles into the environment in order to infect a new host and repeat the cycle. Structural information on ssRNA Lys proteins is only limited to A_2_ of Qβ, a PG-targeting Lys. Even more mysterious is the Lys of the non-PG-targeting group. There currently remains a lack of structural information and knowledge of the mechanisms of this group. 

In summary, there are still unexplored structural ventures that could potentially help in understanding and capturing the life cycle of ssRNA phages in situ, especially for non-*E. coli* ssRNA phages. Additionally, it is important to explore metagenome sequences to identify hosts for non-culturable ssRNA phages, which could potentially become tangible with more structural information.

## Figures and Tables

**Figure 1 viruses-15-01985-f001:**
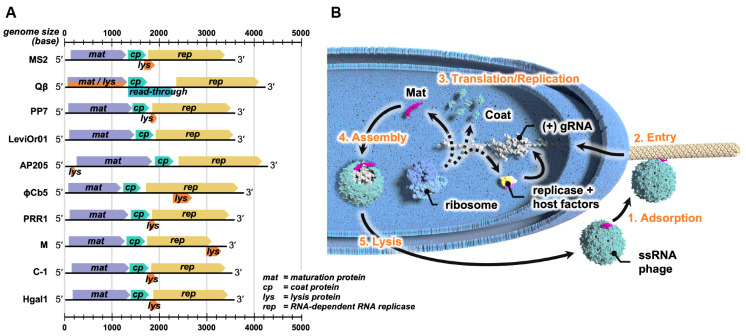
The genome architecture and the infection cycle of ssRNA phages. (**A**) The genome architecture of some culturable ssRNA phages. Their genomes contain three core genes: *mat* (encoding maturation protein), *cp* (encoding coat protein)*,* and *rep* (encoding β-subunit of the replicase). (**B**) The infection cycle of ssRNA phages. The cycle starts with (1) the adsorption of ssRNA phage to the side of the retractile pilus; (2) the retraction of the pilus promotes the entry of gRNA into the host cell; (3) the genome is translated by host ribosomes to manufacture the viral structural proteins, as well as the replicase, which with host factors, synthesizes more of the viral genome; (4) the gRNA is encapsidated by CPs and interacts with the Mat to assemble into mature virions; and (5) the Lys encoded in the gRNA of ssRNA phages induces host lysis to release new viral progeny.

**Figure 3 viruses-15-01985-f003:**
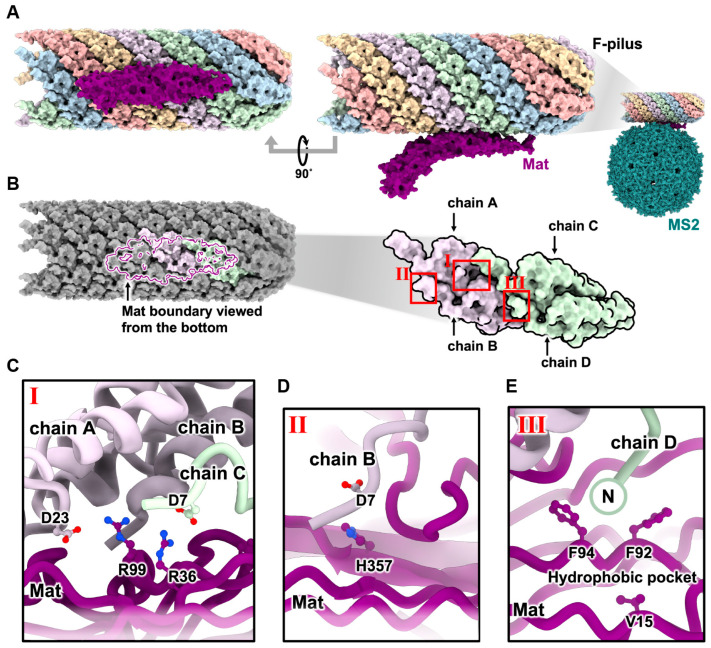
The interaction of MS2 and the F-pilus [31]. (**A**) The interaction of Mat^MS2^ (purple) and F-pilus, represented with five different colors for each helical strand. (**B**) Zoom-in view of panel A focusing on the binding site where Mat^MS2^, shown as a purple boundary, is interacting with four pilin subunits. (**C**–**E**) Zoom-in views from panel B, denoted by roman numerals from I to III, rotated 90° to illustrate the reported contacts between Mat^MS2^ and pilin subunits.

**Figure 4 viruses-15-01985-f004:**
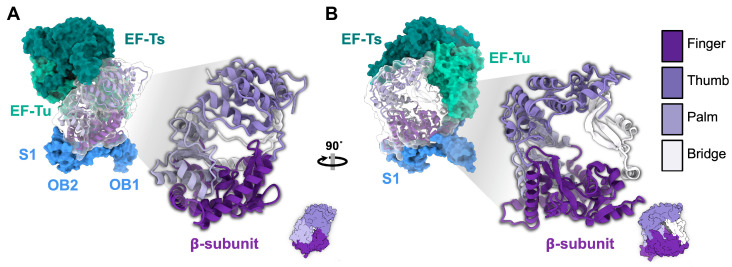
The Rep^Qβ^ holoenzyme complex (PDB 4R71). (**A**) The β-subunit of the replicase (violet) forms a holoenzyme complex by hijacking the host factors: EF-Ts (dark green), EF-Tu (light green), and the ribosomal protein: S1 (blue). S1 is only shown with OB1 and OB2 domains, which interact with the β-subunit. (**B**) The 90° rotation of the model from panel A. The ribbon model of β-subunit which contains a four-domain architecture: finger, thumb, palm, and bridge (illustrated using different shades of purple/violet and white).

**Figure 5 viruses-15-01985-f005:**
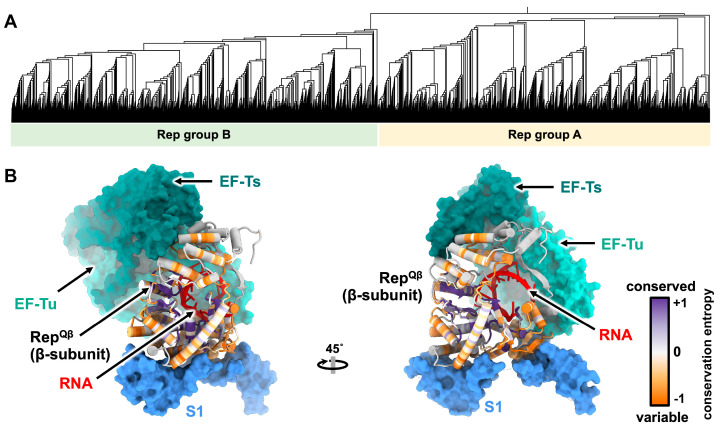
The phylogenetic tree and conservation among ssRNA phage replicases. (**A**) The phylogenetic tree of 15,452 ssRNA phage sequences with 90% average nucleotide identity from all available ssRNA phage sequences shows two distinct clades of ssRNA phages based on two distinct Rep groups [3]. The phylogenetic tree generated is based on the supplementary data in [4]. (**B**) The holoenzyme replicase complex of Qβ (PDB 4R71), bound to RNA (PDB 3BSN). The β-subunit of the replicase is colored by conservation from MUSCLE alignment of 149 sequences from pre-2018 ssRNA phages. The conservation plot was created using ChimeraX software based on conservation entropy value (AL2CO) calculated by ChimeraX [47,48].

**Figure 6 viruses-15-01985-f006:**
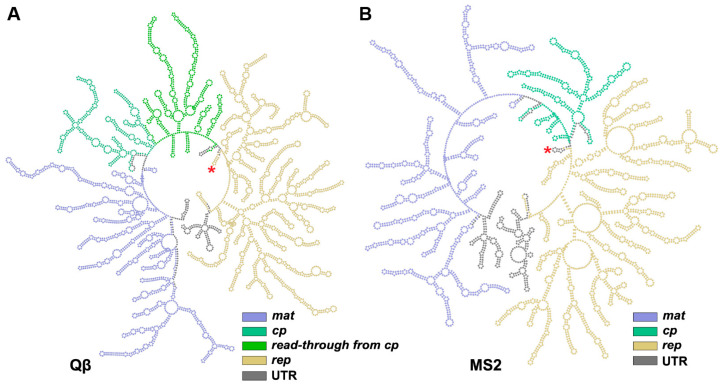
The secondary structures of gRNA from (**A**) Qβ [56] and (**B**) MS2 [20,29]. The genomes were colored based on open-reading frames encoded. The *lys*^MS2^ gene was not annotated because it overlaps both *cp* and *rep*. The *lys*^Qβ^ is its own *mat*. The untranslated regions (UTR) are colored grey. The operator, indicated by red asterisk, folds into the RNA stem containing 3′ end of the preceding UTR and 5′ end of the *rep* gene.

**Figure 7 viruses-15-01985-f007:**
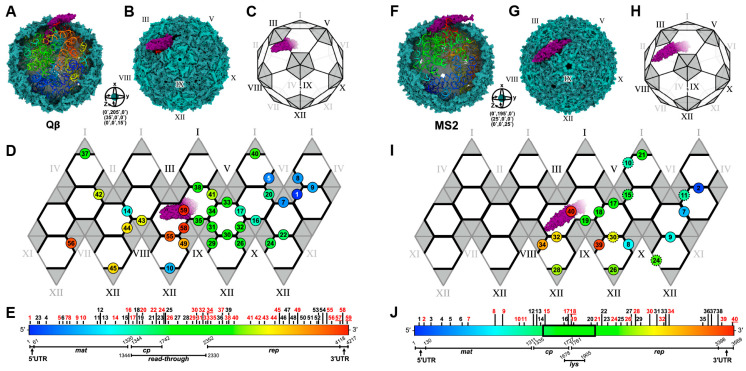
The genome packaging and CP/RNA stem–loop interactions of Qβ (**A**–**E**) and MS2 (**F**–**J**). (**A**) The atomic model of Qβ (PDB 7LHD) with the capsid (green) partially removed to reveal gRNA (rainbow). (**B**) The model from panel A illustrated at a different rotational angle around the XYZ axes, as indicated in the inset. (**C**) A cartoon representation of the model in panel (**B**), showing the pentameric faces (gray) and hexametric faces (white) of the Qβ capsid. Each pentameric face was labelled with roman numerals from I to XII to establish their locations on the capsid. The roman numerals, shown in gray, represent the faces that point inwards to the paper, while the ones shown in black represent those that point outwards. (**D**) The planar representation of the icosahedral surface from panel (**C**) shows the CP dimers (black line) that interact with “operator-like” RNA stem-loops. The RNA stem-loops are shown with numbers corresponding to those labelled in panel (**E**) [56]. (**E**) The gRNA sequence of Qβ shown in rainbow to match those in panels (**A**,**D**). The red numbers indicate operator-like RNA stem-loops plotted in panel (**D**), while the black numbers are non-operator-like RNA stem-loops. (**F**) The atomic model of MS2 (combined PDB from 6NM5, 5MSF and RNA modeled using conformation 1 from Chang et al. [29]) represented as in panel A. (**G**) The atomic model from panel (**F**) rotated around the XYZ axes, as indicated in the inset. (**H**) The cartoon representation of panel (**F**), as shown in panel (**C**). (**I**) The planar representation of the icosahedral surface from panel (**H**) showing the MS2 gRNA stem-loops interacting with capsid, represented in the same way as in panel (**D**). The solid black-outline circles are the stem-loops identified by Dai et al. [20], while the black dotted-outline circles were the additional stem-loops identified in gRNA Conformation 1 by Chang et al. [29]. (**J**) The gRNA sequence of MS2 is shown in a rainbow to match those in panel (**F**,**I**). The region marked with black box is the region reported to be a flexible region by Dai et al. [20].

**Figure 8 viruses-15-01985-f008:**
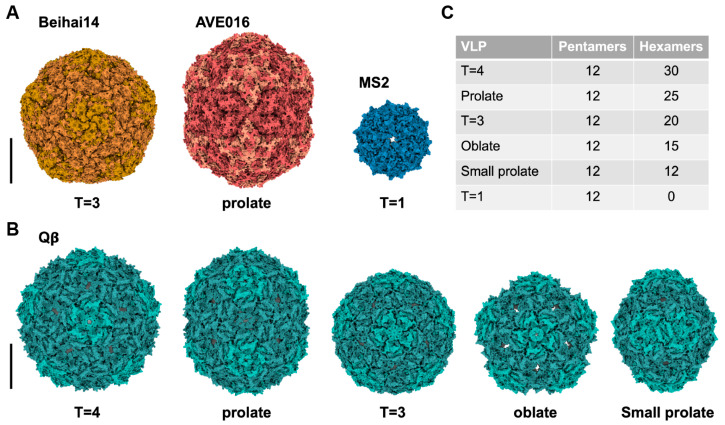
Examples of different forms of VLPs observed in ssRNA phages. (**A**) The VLPs observed from overexpression and/or mutagenesis of CPs. The Beihai14 (PDB 6YFD) and AVE016 (PDB 6YFB) are non-culturable ssRNA phages with their CPs recombinantly expressed and forming a large T = 3 and prolate particles, respectively. The S37P mutation of CP^MS2^ resulted in T = 1 particle (PDB 4ZOR) (**B**) The VLPs of Qβ, which were observed through wild-type infection (PDB 5VLY, 7LGE, 7LGF, 7LGG, and 7LGH). (**C**) The table illustrates the number of pentamers and hexamers for each VLP form. Scale bars in this figure denote 100 Å.

**Figure 9 viruses-15-01985-f009:**
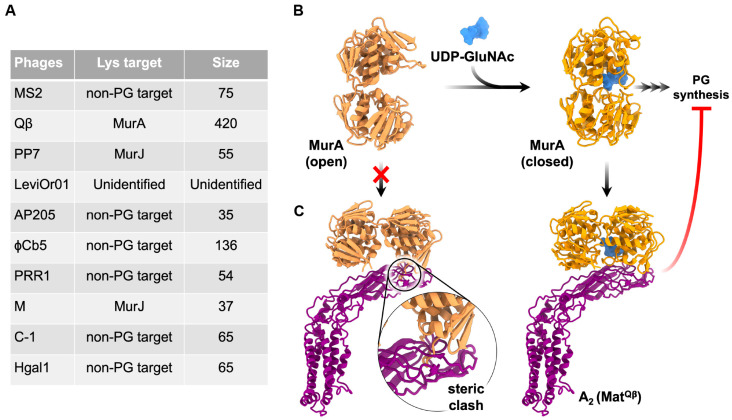
Cell lysis induced by Lys of ssRNA phages with the structural mechanism only revealed for Qβ. (**A**) Single lysis proteins from previously known ssRNA phages, with the name, target and size in amino acid length listed. (**B**) The MurA in its open conformation (PDB 1EJD) without its substrate, UDP-GluNAc. Upon the substrate binding, MurA changes to a closed conformation (PDB 3VCY). The released product will be further used by downstream enzymes for peptidoglycan biogenesis. (**C**) The open conformation of MurA was unable to bind to A_2_ (PDB 5VM7) due to the steric clashes between the loops from MurA and the tip of Mat β-region. In the closed conformation, the MurA/UDP-GluNAc complex can bind to A_2_, inhibiting MurA function.

**Table 2 viruses-15-01985-t002:** Mature virion percentages of ssRNA phages from published cryo-EM data.

Phages	Total Particles	Mature Virion Particles *	Reference
MS2	22,441	18,977 (~85%)	Konning R.I., 2016 [22]
	48,276	44,897 (~93%)	Meng R., 2019 [31]
Qβ	51,815	12,975 (~25%)	Gorzelnik K.V., 2016 [21]
	248,445	76,843 (~31%)	Cui Z., 2017 [24]

* Mature virion particles are the particles that contain Mat proteins and have defined gRNA density.

## Data Availability

Not applicable; No new data were generated.

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
