# Peer review of "Recent Advances in Structural Studies of Single-Stranded RNA Bacteriophages"

_viruses, 2023, doi:10.3390/v15101985_

Round 1
Reviewer 1 Report
This is a good new review of ssRNA phages that were found nearly half century ago. The review provides a comprehensive summary of studies performed during the last decade demonstrating significant progress in understanding structure-function links in this type of pages.
Some questions to the authors of this review:
The authors write that “ssRNA phages are lytic phages”, therefore their genome is normally not incorporated into genome of a host bacteria. So, it is not quite clear (and the authors did not explain the principles of the search) how thousands (65,814) of new ssRNA genomes were identified. Which data bases were used utilising metagenomic and transcriptomic approaches? How some genome sequences where assigned to the ssRNA phages and their corresponding proteins, while the structural information was known for only ~30 phages (MS2, Qβ, PP7, LeviOr01, ɸCb5, AP205, PRR1, fr, GA plus capsids ~20 of virus like particles of nonculturable phages). Which criterions were used to identify phage proteins?
The authors have written that these 65 thousand genomes allowed for the taxonomic “reclassification of ssRNA phages, resulting in the creation of two new orders, six families, 428 genera, and 882 species”. However, it was not explained which criteria were used in this differentiation and what were the main characteristics of orders and families. The information on that would make this review more valuable.
The authors will improve the citation index of their review if they well explain to a reader what they mean as HMM profile (Hidden Markov Models) and what its role in sequence analysis. The authors should not go into the mathematical explanation of such analysis but should explain why the assessment of HMM sequence profiles will provide an important information (on what?). Moreover, possibly methods of the deep learning like AlphaFold 2, Rossetta, I-Tasser, ESMFold provide significantly more complete information nowadays and results of this kind of analysis have to be cited.
It would be good if the authors will provide more information on the Maturation protein. They have demonstrated the maturation protein structures of only two phages: MS2 and Qβ. While these maturation proteins have similar folds of two domains that compose them, these domains have different orientations between each other in these two phages. Was it important or not? Does the maturation proteins act as a switch for the release of ssRNA into pilis? It was interesting to see that 5’ RHA end is attached to the maturation protein and, apparently, it was the last end of the genome packed into the capsid and will be the first one released into the host cell. Do the maturation proteins form a channel for the entrance and release of the phage gRNA? Was the predicted by AlphaFold structure of 6NM5 with the atomic model of the β-sheet region confirmed by structural analysis?
A theory that the fibres to which the virus became adsorbed should be detached from the bacterium sounds a bit confusing: how then the virus will infect the bacterium since it should go away together with the fibre to which the viral particle is attached? Something has been missed in this part of the text.
“These studies provided valuable insight into the binding of MS2 to F-pili and resolved the longstanding debate on whether the genomic RNA (gRNA) is ejected upon binding “ (lines 147-148). The results of the study mentioned here have to be explained and the evidence should be given here as well. The statement that a “valuable insight” was provided is not very informative. The ssRNA should be injected into the host cell, otherwise new phages will not emerge out of the bacteria.
“These results revealed the gRNA packing preference in mature virions (Lines 296-298)”. A common and accepted idea is, that virions became mature only after packaging of the genome into the capsid. The process of packaging is often named as a maturation process. Typically, a point of entrance to the capsid became the point of exit for the genome. However, it is not clear so far how the virus maturation takes place in ssRNA viruses: is it a genome that collapsed into a tight knot around which the capsid will be assembled, or the genome will be pumped inside of the preassembled capsid. The authors write that no such elements present in the ssRNA genome which could work as energisers that will work as pumping motor. Could the authors make this paragraph (lines 144-149) a bit clearer?
“While TraD is required for translocation of MS2 gRNA across the cell envelope into the host cytoplasm, it is not essential for pilus detachment“ (Lines 184-186). It was confusing: does the TraD protein is a bacterial protein or it is produced by the phage? If it is related to the bacterium then it works as a trigger for the activating the maturation protein that induces the phage gRNA release, but the pili should be attached to the bacterium working as a channel for the gRNA deliver into the cell.
Minor comments:
Lines 170-175. Do the panels C,D,E in figure 3 show the orientation not from the top as in B but rotated on 90 degrees with the respect long axis of F-pilus? These panels are very confusing.
Line 194. “This viral subunit utilises…” It seems that the authors want to say:“this viral particle…” Usage the term “subunit” in not quite correct, “particle” would be better.
Line 203. The panel A (Figure 4) should be replaced by a panel from Figure 5 B, the left one, since “EF-Tu (light green)“ is not seen at all.
Lines 228-231. “The HMM search on ssRNA sequences revealed ssRNA phage Rep proteins are relatively conserved and can be classified into two clusters (Figure 5A)”. Both figure 5 and its legend are confusing: it is unclear why the authors made two clusters but not 3 or 5. Which criteria were used? What was used in the original paper? How these clusters can be distinguished?
The differences in interactions impossible to see, where the atomic models just rotated in 90 degrees? Areas of interactions are hidden.
Lines 304-330. Figure 7 legend. Panels D and I: they are not “unwrapped capsid maps ” but planar representations of icosahedral surfaces of capsids where the points of interactions of CP with the genome are indicated. It would be good if the major content of the figure legend will be moved to the main text with explanations of the red and black numbers in Figure 7, E and J. How this numbering has been chosen and what do they correspond to.
Line 357. “RNA stem-loops which act as a packaging signal “please explain more clearly, what the authors mean: how a stem-loop can act as a signal? Do the authors mean that some interaction (between which structural elements or components?) induce some conformational changes that will induce that signal?
Line 390, “an arbitrary TTA (which codes for leucine) “. It seems that ATT is not arbitrary but well defined. The lines 367-393 is not easy to follow.
Lines 415-416. Here is the same question related to the very first question on the first page: how the phage genomes could be identified within bacterial genomes. That was not explained.
Line 429. “With an increase in ssRNA phage sequences “. Please rephrase: the authors were thinking of the increasing in the numbers of known sequences, but not of their length.
It would be good ig English will be checked and corrected.
Reviewer 2 Report
This is a really nice review of the status of studies into ssRNA phages. It's timely and well written and I found it very informative. I found it hard to find very much to complain about. I'll just note a few places where it took me some extra time to figure out what was being said. I don't feel very strongly about any of these minor criticisms.
You have interchangeably used the jargon "Mats" to mean Mat proteins. That sent me back through the paper looking for whether I had missed something called Mats.
Line 136 is awkwardly phrased, although it does become clear as you expound further.
Line 148: "..resolved the longstanding debate on whether the genomic RNA (gRNA) is ejected upon binding [35, 36]." You could be a little more clear about what the debate was about and which development you consider to have resolved it. Are references 35 and 36 supposed to represent the outdated view? Is interaction with TraD supposed to be the resolution?
Line 176 could use a clarification of what "pilus detachment" means. As I understand it, it means broken fragments of pili appearing in the culture medium, not the virus detaching from the pilus.
Fig 5A. I'm having a hard time finding a tree figure that looks like that in the cited paper. If it's some .tre file that they had as a supplement, please name it.
Fig. 6. In the addition to the color coding in the color key, there is some other color (or is it black) at the ends of the genome and in a hairpin before the rep gene. Is that untranslated sequence? Is the black? hairpin before rep the "operator" you talk about? Are all the "operators" you mention in untranslated regions?
Reviewer 3 Report
In this review the authors thoroughly outline the structural components of ssRNA bacteriophages. ssRNA phages have held an important place in molecular biology due to important structural and genomic studies, yet very few have been characterized until recent metagenomic data has increased the number of these phages exponentially. The authors do an excellent job of using the framework of the lifecycle of ssRNA bacteriophages to illustrate how structural insights have illuminated various steps of this lifecycle. The information provided is thorough but not overly detailed. Furthermore, the figures presented in the review do a very good job of summarizing and presenting the structural information that has been gathered about ssRNA bacteriophages. The writing is clear and little to no editing is required. Overall the review is excellent and is suitable for publication.
Minor Consideration:
Since much of the difficulty identifying new ssRNA phages has been due to low sequence homology, it would strengthen the manuscript to comment on and/or demonstrate the potential of protein prediction software and the large structural databases that have been generated from metagenomic to identify new ssRNA phages (alphafold2 and esmfold). It seems that this could be a logical step to expanding the known collection of ssRNA phages even further.
